# Is a Maximal Strength-Training Program Effective on Physical Fitness, Injury Incidence, and Injury Burden in Semi-Professional Soccer Players? A Randomized Controlled Trial

**DOI:** 10.3390/healthcare11243195

**Published:** 2023-12-18

**Authors:** Roberto Durán-Custodio, Daniel Castillo, Javier Raya-González, Javier Yanci

**Affiliations:** 1Faculty of Education and Sport, University of the Basque Country (UPV/EHU), 01006 Vitoria-Gasteiz, Spain; punyos3@yahoo.es; 2Faculty of Health Sciences, International University of La Rioja (UNIR), 26006 Logroño, Spain; 3Valoración del Rendimiento Deportivo, Actividad Física y Salud y Lesiones Deportivas (REDAFLED), Department of Didactics of Musical, Plastic and Corporal Expression, Faculty of Education, University of Valladolid, 42004 Soria, Spain; 4Faculty of Sport Sciences, University of Extremadura, 10003 Cáceres, Spain; rayagonzalezjavier@gmail.com; 5Society, Sports and Physical Exercise Research Group (GIKAFIT), Department of Physical Education and Sport, Faculty of Education and Sport, University of the Basque Country (UPV/EHU), 01006 Vitoria-Gasteiz, Spain; javier.yanci@ehu.eus

**Keywords:** football, prevention, performance, injuries, health

## Abstract

The aims of the study were to analyze the effects of a 12-week maximal strength- training program on injury incidence, injury burden, and physical fitness in semi-professional soccer players and to compare the perceived exertion load and well-being state between injured and non-injured soccer players. Twenty semi-professional male soccer players participated in this study. Participants were randomly allocated to an experimental group (EG, *n* = 10 players), who performed a maximal strength-training program, or to a control group (CG, *n* = 10 players), who only performed their regular soccer training. Physical fitness was measured at baseline and after the training program. In addition, the injury incidence, burden, training/match load, and the state of well-being of the players were recorded. The EG showed significant improvements in vertical jumps, change in direction ability, linear sprints, repeated sprint ability, isometric strength (*p* < 0.003; effect size = 1.78–11.86), and quadriceps–hamstring imbalance in both legs (*p* < 0.001; effect size = 2.37–3.71) in comparison to the CG. In addition, the EG players showed a significantly (*p* < 0.05) lower injury burden (*p* < 0.001, relative risk = 5.05, 95% confidence interval = 3.27–7.79). This study demonstrated the beneficial effects of a 12-week maximal strength-training program on physical fitness attributes and injury burden in semi-professional soccer players.

## 1. Introduction

Soccer is a very demanding team sport in which players must be in optimum physical condition to perform satisfactorily in every phase of the season [1]. The high physical and physiological responses predispose players to a high risk of injury, resulting in a multitude of injuries [2]. Primary concerns arising from injuries include acute or chronic health disruptions and the inability to compete during the injury period [2,3]. This fact may imply the loss of several official matches depending on the type and severity of the injury [4]. In addition, being injured can also lead to low self-esteem, pessimism, hopelessness, concentration difficulties, excessive self-criticism, anxiety, and depression [5]. Injuries are also associated with high economic costs, derived from the salary received by the player during the period of absence and from the medical, surgical, and rehabilitation treatments [6,7]. Due to all of these aspects, injuries have become a relevant investigation topic [4,8,9,10]. It has been reported that the injury rate in professional soccer is 8.1 to 14.3 injuries per 1000 h of exposure [11,12,13]. Moreover, aside from injury incidence, the concept of ‘injury burden’ has been used as a pivotal indicator of injury severity [4,14]. Several previous studies have elucidated that severe injuries (causing absences exceeding 28 days) constitute between 16% and 22% of all injuries in professional soccer players [2,11]. The most common subtypes of severe injuries included hamstring strains (12%), anterior cruciate ligament injuries (9%), quadriceps strains (7%), and groin pain/strains (6%) [2]. Consequently, an average team at this level can anticipate eight severe injuries per season [2]. Regarding semi-professional soccer players, some authors have noted injury rates ranging from 6.2 to 12.4 injuries per 1000 h of exposure [15,16], with severe injuries accounting for 12% to 18% of cases [15,16]. Currently, a greater number of studies have been conducted in professional soccer, but it would be interesting to extend the research to analyze the context of injuries in semi-professional players. Therefore, it could be interesting for strength specialists and physical trainers to find training strategies for reducing the risk of injury in soccer players.

Aiming to reduce the number and severity of injuries, the scientific literature has demonstrated a high interest in evaluating the efficacy of training strategies for injury prevention [10,17,18,19,20,21,22]. In this regard, several research studies have revealed the potential of strength training to mitigate the severity of both muscle and joint injuries [17,21,23,24]. This could be due to the enhancement of functional properties and the joint-stabilizing function derived from strength-training programs, factors that may prove instrumental in safeguarding against sports-related injuries in soccer players [25]. Recent emphasis has been placed on the significance of lumbo-pelvic muscle function in the prevention and treatment of hamstring injuries. It has been observed that the activation level and strength of the gluteus medius during high-speed running exhibit a strong correlation with hamstring injuries [26]. Furthermore, it has been shown that greater leg asymmetries are predominantly localized in the gluteal muscles [27]. These findings suggest that the hamstring and gluteal muscles may play crucial roles in strength training for soccer players, potentially averting injuries and enhancing physical performance [26,27]. In this sense, some studies have included maximal strength training consisting of a low number of repetitions (1–5) with high loads (85–100% of one maximum repetition, 1 MR) in order to reduce the injury risk, improve sprint ability, decrease fatigue in repeated sprints, and increase muscle fascicle length [28,29,30,31]. Specifically, the maximal strength training could be beneficial for improving neural adaptation and strength gains without increasing the body mass [9,29]. Despite the benefits of the maximal strength-training programs, there is a gap in the scientific knowledge regarding the impact of this type of strength training on the incidence and severity of injuries in soccer.

Due to the need for optimal physical conditioning for increased soccer performance, in addition to understanding the potential effects of an intervention program on soccer injuries, previous research has focused on analyzing the influence of training/match load on injury risk [4,14,32,33]. Some researchers have demonstrated that training/match loads are associated with injury risk, exposing players to potentially injurious situations [33,34]. Specifically, these studies observed that a higher rate of perceived exertion load was correlated with an increased injury risk [35,36]. In this context, it may be pertinent in studies to investigate the effects of a training program on injury incidence and burden. In addition, multiple factors believed to contribute to athletic injuries, such as ‘chance’, foul play, environmental factors, fatigue, warm-up, and style of play, must be considered [28]. Furthermore, it has been reported that changes in the physical condition of soccer players over time can affect injuries [8]. Some authors have explained the suitability of maximal strength training to increase sports performance and reduce the number and severity of injuries [37,38]. Training loads and changes in the physical condition of semi-professional players tend to be less controlled, primarily due to limited resources [15,16]. Furthermore, specifically in non-professional or semi-professional soccer, players have to combine the participation in this sport with other occupational activities, so it would be relevant to know how players face the daily training and competition (i.e., well-being state). In non-professional soccer, there have been several investigations related to the effects and benefits of strength training in injury prevention [39,40]. Therefore, this study could help to corroborate this information and complement it in a novel way with new parameters such as the influence of training load and players’ state of well-being. 

Knowing the benefits and detriments of strength-training programs on physical fitness and injury risk would seem to be essential for making proper decisions when implementing such programs. This study aimed to analyze the effects of a 12-week maximal strength-training program on injury incidence, injury burden, and physical fitness variables in semi-professional soccer players and also to compare the RPE_L_ and state of well-being between injured and non-injured players.

## 2. Materials and Methods

### 2.1. Experimental Design

A randomized controlled trial was conducted as a descriptive pilot study to analyze the effects of 12 weeks of maximal strength training on physical fitness attributes and lower-limb injury incidence and injury burden and also to compare the RPE_L_ and well-being among injured and non-injured soccer players. The tests used at the beginning (baseline) and after the training intervention (post) were vertical jumps (counter movement jump, CMJ; squat jump, SJ), change in direction ability (505-CODA), linear sprints (10, 20, and 40 m), repeated sprint ability (RSA, 5 × 30 m), and isometric strength exercises in the quadriceps, hamstrings, hip abductors, and hip adductor muscle groups in both dominant and non-dominant limbs. In addition, injury incidence and burden were recorded during the total period of the study and the training/match load was quantified by RPE_L_. Also, Hooper’s wellness status was recorded to know how the players faced the training and competition. In this sense, jump testing (i.e., CMJ and SJ) and isometric strength were measured in a performance lab (18 °C, 60–70% relative humidity), whereas 505-CODA, sprinting, and RSA were assessed on an artificial grass field where the team performed their usual training sessions and players wore their own soccer boots. All tests were carried out in the morning between 10.00 a.m. and 12.00 p.m. Likewise, players were instructed to take their last meal 3 h before the beginning of the tests, not to drink any caffeinated beverages, and not to perform intense physical exercise. A strength and conditioning specialist supervised the tests and gave verbal encouragement during all of the protocols [20].

### 2.2. Participants

Twenty semi-professional male soccer players participated in the study. Soccer players were randomly assigned to a control group (CG; *n* = 10, age = 25.8 ± 2.0 years, height = 178.3 ± 3.6 cm, body mass = 74.7 ± 5.7 kg, body mass index [BMI] = 23.5 ± 1.0 kg/m^−2^), who did not perform additional specific maximal strength training, and to an experimental group (EG; *n* = 10, age = 24.5 ± 3.1 years, height = 176.6 ± 4.4 cm, body mass = 72.8 ± 6.6 kg, BMI = 23.3 ± 1.6 kg/m^−2^), who performed maximal strength training in addition to their usual training. The number of participants in each group was based on previous literature [41,42]. The weekly training volume was four soccer-specific training sessions of approximately 105 min each (totalling 420 min) and one official match on a weekend. Players were included in the study if they performed strength training for at least 2 years before participating in the study and had no pre-study neuromuscular injury. Players were excluded if they did not participate in at least 80% of the training sessions (i.e., soccer and strength sessions) over the 12-week period. The two goalkeepers were excluded from statistical analysis due to their special role in soccer practice but players who were injured at the beginning of the investigation were also excluded (n = 2). All participants were informed of the benefits, procedures, and potential risks of the study and they gave their written informed consent to participate. The study was conducted in accordance with the Declaration of Helsinki (2013), and the protocol was approved by the ethics committee of the Valladolid East Health Area (Code PI 22-2793 NO HCUV).

### 2.3. Procedures

The EG players conducted a 12-week mid-season intervention during the months of February to April, in addition to their soccer training. This program was based on the progression of four exercises, following a maximal strength orientation (i.e., 85–95% of 1 MR, 3–4 repetitions, 3 min rest between sets). The strength sessions were performed on two alternate days at the same time of day (Tuesday and Thursday at 11.00 a.m.), with a duration of 35–40 min, and were applied after the soccer-specific drills [43,44]. Wellness was assessed using Hooper’s wellness questionnaire around 1 h before the training and competition. Ten minutes after the match or training sessions, the players were asked to declare their RPE. The baseline and post-intervention tests were performed on three alternate days during the same week. On the first day, the 505-CODA and linear sprints were performed. After a rest of 48 h (second day), the RSA tests were performed. Finally, on the third day, after 48 h of rest, the isometric strength exercises in hamstring dominant (ISOHAMSd) and non-dominant (ISOHAMSnd), quadriceps dominant (ISOQUAd) and non-dominant (ISOHAMSnd), abductor dominant (ISOABDd) and non-dominant (ISOABDnd), and adductor dominant (ISOADDd) and non-dominant (ISOADDnd) limbs were performed. For the jump and speed tests, a specific warm-up was performed using running and articular mobility exercises with a duration of 10 min. For the isometric strength tests, a specific 15 min warm-up was performed by executing the four exercises that were developed in the test with an intensity of 40–60% of 1 MR, performing two series of each exercise (between 10 and 15 repetitions) with a rest of 30–60 s between series.

### 2.4. Intervention Program

A total of 24 strength training sessions were completed before proceeding to the post-tests ensuring a minimum of two rest days between sessions. Previous to each strength session, a specific warm-up was completed in which the four exercises performed in the study were executed, completing two sets of each exercise at an intensity of 40–60% of 1 MR, with 30–60-s rest periods. The selected exercises were: unilateral horizontal leg press; unilateral lateral leg press with 45° inclination of the supporting foot with respect to the surface; knee extension; and knee flexion (Figure 1). All exercises were performed on inertial devices (Technogym^TM^ and Precor^TM^, Madrid, Spain) as circuit training at 85–95% intensity of 1 MR, with three series of each exercise, performing 3–4 repetitions and with 3 min of recovery between series, completing a total of 12 series. In Weeks 1–2 (W1–2) the training was at an intensity of 85% of 1 MR and 4 repetitions were performed; in W3–4, 90% of 1 MR and 4 repetitions; and in W5–12, 95% of 1 MR and 3 repetitions (Table 1). To calculate the 1 RM, a submaximal and modified protocol was performed [43]. First, the player selected a weight to perform 8–10 repetitions, allowing 3 min for recovery. Then the player increased the load to perform one set of 5–6 repetitions, with 5 min of rest. Finally, the player increased the load to perform one set of 3–4 repetitions. An estimated calculation of 1 MR was then made using the formula proposed by O‘Connor et al. [45] (1 MR = Weight lifted in kilograms × [1 + 0.025 × *n* repetitions]) and the corresponding training percentages were calculated for each player.

### 2.5. Perceived Exertion Load (RPE_L_)

The training/match load was measured by the RPE_L_ over the 12-week period according to the method proposed by Foster [46]. Players were asked to state their RPE on a scale of 0–10 [46] individually 10 min after training and matches, without the presence of other counterparts. This value is multiplied by the training or competition duration to determine the workload (RPE_L_) [47]. The players were familiarized with the use of the RPE scale during the training and match sessions in the preseason period.

### 2.6. Wellbeing State

The Hooper questionnaire [48] was applied to assess how players felt before training and competition. The questionnaire was administered individually without the presence of other players. Every player declared their feelings individually and chose a number from 1 (very very high) to 7 (very very low) for Hooper’s sleep item and from 1 (very very low) to 7 (very very high) for the remaining items (fatigue, stress, and muscle soreness) on Hooper’s questionnaire [48]. The Hooper index was calculated as the sums of scores of the four Hooper questionnaire items: sleep, fatigue, muscle soreness, and stress [48].

### 2.7. Physical Fitness Attributes

#### 2.7.1. Vertical Jump Performance

Soccer players carried out a bilateral CMJ, a CMJ with the dominant (CMJd) and non-dominant leg (CMJnd), and a bilateral SJ [49]. The selection criteria for determining the dominant leg were based on each player’s soccer ability (i.e., hitting leg) [50]. The CMJ was performed by a flexion-extension of the hips and knees at the highest velocity, with a maximum knee flexion angle of 90° and with the help of arms swinging to reach the maximum possible height. The SJ was performed with hands on the hips and the trunk straight and then a maximum vertical jump was executed, starting from the 90° knee flexion position, without any rebound or countermovement. For all jumps, two attempts were performed and the best one was selected for further analysis. The jumps were performed with a 1 min rest between attempts and all players were verbally encouraged to perform the highest possible jump. A contact platform (Optojump Next, Microgate^TM^, Bolzano, Italy) was used to measure jump height (cm), calculated as *h* = *gt*^2^/8, where *h* = height (cm), *g* = acceleration due to gravity (9.81 m·s^−2^), and *t* = flight time (s) of the jump [51]. The leg asymmetry of the CMJ (CMJLA) was calculated using the formula: CMJLA (%) = (Dominant − Non-dominant)/Dominant × 100) [49]. In addition, the elastic index (EI) was calculated as the difference in percentage (%) between CMJ and SJ using the formula: EI (%) = (CMJ − SJ) × 100/SJ) [52]. Intraclass correlation coefficient (ICC) values for CMJ, CMJd, CMJnd, and SJ were in the range 0.906–0.986.

#### 2.7.2. Change in Direction Ability (CODA)

The players performed the 505-CODA test, which consisted of an acceleration run (to increase speed) from the line marked on the ground to the first marker placed at 10 m, then, once past the first marker, a 5 m sprint to the second marker (placed 15 m from the line marked on the ground) and a 180° change in direction, followed by a 5 m sprint back past the first marker (placed 10 m from the line marked on the ground) [53]. A photocell (Polifemo, Microgate^TM^, Bolzano, Italy) located over the start/finish line was used to record the time. Two attempts were made to turn with each leg (dominant: 505-CODAd; non-dominant: 505-CODAnd) and the best one was selected. A 2 min rest was left between attempts. The leg asymmetry of the 505-CODA (505-CODA_LA_) was calculated using the formula: 505-CODA_LA_ (%) = (Dominant − Non-dominant) / Dominant × 100). The ICC values were 0.858–0.878 for all CODA tests.

#### 2.7.3. Linear Test Sprints

Soccer players performed a 40 m linear sprint [54], with splits at 10 (SPR10), 20 (SPR20), and 40 m (SPR40) distances [54]. The starting point of the sprint was 0.5 m before the start. Three attempts were performed and the lowest time was selected. A 4 min rest was allowed between sprints and all players were verbally encouraged to achieve the best possible time. Four photoelectric cells (Polifemo, Microgate^TM^, Bolzano, Italy) were used for the measurement in this test. ICC values were 0.930–0.968 for all linear sprint tests.

#### 2.7.4. Repeated Sprint Ability (RSA 5 × 30 m)

Each player performed five sprints of 30 m at maximum speed with 25 s of recovery between each sprint [55]. The sprint starting point was 0.5 m before the start. The sprint time was measured using two photoelectric cells (Polifemo, Microgate^TM^, Bolzano, Italy) set at 0 m and 30 m distances. The RSAtotal was calculated as the sum of the time of the five sprints. A fatigue index (RSASdec), used in previous studies [55], was also calculated with the following formula: RSASdec = [(RSAtotal/(RSAmin × 5) × 100] − 100) [56].

#### 2.7.5. Isometric Strength

Soccer players performed isometric strength contractions during 5 s for the following muscle groups: quadriceps, hamstrings, hip abductors, and hip adductors. The strength was measured using a dynamometer (Carp Spirit Water Queen Digital Scale 50, BIODEX System Pro 4™, System 4 advance v.4.2, New York, NY, USA), as previously validated [57]. For ISOHAMSd and ISOHAMSnd, the players were seated on the stretcher, the knee was fixed and with 90° of flexion, and then knee flexion was performed against the dynamometric tape (Figure 2A). For ISOQUAd and ISOQUAnd, the players were seated on the stretcher, the knee was completely fixed and with 90° of flexion, and then knee extension was performed against the dynamometric tape (Figure 2B). For ISOABDd, ISOABDnd, ISOADDd, and ISOADDnd, the players were supine, with the knee in full extension and the hip neutral, and then hip abduction and adduction were performed against the dynamometric tape, keeping the knee fully extended (Figure 2C,D). Participants performed two attempts with each leg and the best result was selected for further analysis. A 1 min rest was taken between attempts and all players were verbally encouraged to execute the tests with maximum force. The leg asymmetry of each isometric test (ISOQUA_LA_, ISOHAMS_LA_, ISOABD_LA_, and ISOADD_LA_) and the imbalance between agonists and antagonists (Q–H imbalance and Abd–Add imbalance) were also calculated, using the following respective formulae: LA (%) = (Dominant − Non-dominant)/Dominant × 100) and Imbalance (%) = (Agonist − Antagonist) × 100/Agonist) [49,58]. ICC values were 0.994–0.997 for all isometric strength tests.

### 2.8. Injuries

During the 12-week intervention period, the number of injuries, mechanism of injury, type of injury, body region, muscle structure, time of injury, time of absence from training or match, and lower extremity load were recorded. The criteria of the Union of European Football Associations (UEFA) model [11] were followed for this purpose. Lower extremity injuries were diagnosed by the medical staff and recorded by the physical trainer for the study. Treatment and the recovery follow-up were also carried out by the medical staff. Injury was defined as “an injury occurring during a scheduled training session or match that caused absence from the next training session or match” [11]. Subsequently, the injury incidence (injuries/1000 h exposure) and injury burden (days of absence/1000 h exposure), defined as “the number of days lost per 1000 h of exposure” [4] were calculated. The criteria used to consider exposure were as follows: “the time (in hours), both training and match, during which the player is in a position to sustain an injury, and the incidence rate refers to the number of injuries sustained during practice, both in training and competition, per 1000 h of exposure” [24]. Specifically, the incidence rate was considered in this study [59]. A player was considered fully recovered after an injury when the medical staff cleared him to fully participate in team training and matches [60].

### 2.9. Statistical Analysis

Data are presented as the mean ± standard deviations (SD). The Shapiro–Wilk test and Levene test were conducted to assess the normality of data distribution and the homogeneity of variances, respectively. Data related to lower extremity injuries (i.e., incidence and burden) are presented as the number per 1000 h of exposure and the number of absence days/1000 h exposure, each with 95% confidence intervals (CI). The rate ratios (RR) with 95% CI and the Z-test [61] were calculated for between-group differences (i.e., EG and CG) regarding to lower extremity incidence and burden. Between-group differences at pre-intervention were tested using independent t-tests. An analysis of covariance (ANCOVA) was performed to detect possible between-group differences, assuming baseline values as covariates. A paired-samples *t*-test was used to evaluate within-group pre-to-post differences. An independent *t*-test was applied to analyze the differences in RPE_L_ and Hopper variables between EG and CG in each week period (W1–12, W1–6, and W7–12). Also, an independent *t*-test or Welch test was performed to analyze the differences in the same variables between those injured and not injured for all players and for the EG and CG. Cohen’s d effect size (ES) was calculated [62] to examine practical significance. The results obtained were interpreted as small (0.00 ≤ d ≤ 0.49), moderate (0.50 ≤ d ≤ 0.79), and large (d ≥ 0.80). Data analysis was performed using the Statistical Package for the Social Sciences (SPSS^TM^ Inc., version 27.0 for IOS, Chicago, IL, USA). For all the analyses, the significance level was set at *p* < 0.05.

## 3. Results

The total RPE_L_ for the team was 33,568.6 ± 6924.3 AU for the W1–12 period, 17,261.2 ± 3610.3 AU for the W1–6 period, and 16,307.4 ± 4299.1 AU for the W7–12 period. The Hooper index for all the team players was 690.5 ± 150.6 for the W1–12 period, 350.7 ± 79.2 for the W1–6 period, and 339.8 ± 91.0 for the W7–12 period. Hooper’s sleep values for all the team players were 132.7 ± 29.7, 60.6 ± 15.4, and 56.1 ± 17.7 for the W1–12, W1–6, and W7–12 periods, respectively; Hooper’s stress values were 132.7 ± 37.2, 66.4 ± 18.4, and 59.6 ± 15.9, and Hooper’s fatigue values were 210.7 ± 47.4, 106.7 ± 26.5, and 66.2 ± 18.8 for the periods W1–12, W1–6, and W7–12, respectively. On the other hand, Hooper’s muscle soreness values were 230.4 ± 46.9, 117.0 ± 24.5, and 106.1 ± 26.4 for the periods W1–12, W1–6, and W7–12, respectively. Comparisons between the EG and CG in RPE_L_ and Hooper values for each week and the 6-week periods (W1–6 and W7–12) are shown in Table 2. Despite the EG having a higher RPE_L_ in Weeks 3, 7, and 12 compared to the CG (*p* < 0.01 or *p* < 0.05, ES = −1.15 to −1.57, large), no significant differences (*p* > 0.05) were found between the groups overall in the W1–6, W7–12, and W1–12 periods. On the other hand, although the EG reported a higher Hooper index in Week 10 compared to the CG (*p* < 0.01, ES = −1.35, large), no significant differences were found between the groups overall in the W1–6, W7–12, and W1–12 periods. Regarding Hooper’s sleep, stress, and muscle soreness items, no significant differences were observed between the CG and EG in any of the analyzed periods (i.e., W1–12, W1–6, W7–12). Significant differences were observed between the CG and EG in Hooper’s fatigue values in the W1–12 period (188.9 ± 47.2 vs. 232.5 ± 38.2, *p* = 0.036, ES = −1.01, large) and in the W1–6 period (95.1 ± 26.4 vs. 118 ± 22.2, *p* = 0.046, ES = −0.96, large), but no significant differences were observed in the W7–12 period (66.0 ± 22.9 vs. 65.4 ± 14.7, *p* > 0.05, ES = 0.08, small).

The results obtained concerning the number of injuries, mechanism, type, region, time of injury, and time of absence are presented in Table 3. Six players were injured during the intervention period of the study, five in the CG and one in the EG. Soccer players in the CG suffered five musculoskeletal injuries (113 absence days). Five players were injured, which is 50% of the CG (*n* = 10), while EG players suffered one musculoskeletal injury (25 absence days). In the EG, only one player was injured, which is 10% of the EG (*n* = 10) during the 12-week intervention period. Of the total injuries in the intervention period, four occurred in W1–6 (three in the CG and one in the EG) and two in W7–12 (two in the CG and none in the EG).

The injury incidence and burden were 7.31/1000 h exposure and 168.17/1000 h exposure, respectively, for all players. The injury incidence and burden differences between the EG and CG are shown in Figure 3. Even though no significant differences (*p* = 0.120) between the groups in incidence were observed (Figure 3A; CG: 6.45 vs. EG: 1.15 injuries/1000 h of exposure, RR = 5.59, 95% CI = 0.65–47.81), significant differences (*p* < 0.001) in the injury burden were reported (Figure 3B; CG: 145.77 vs. EG: 28.87 absence days/1000 h of exposure, RR = 5.05, 95% CI = 3.27–7.79).

The changes in physical fitness attributes after the 12-week period intervention are displayed in Table 4. There were no baseline differences between groups in any variable. The ANCOVA model revealed between-group differences in most of the physical fitness variables in favor of the EG. The baseline to post-training values improved in most of the physical fitness attributes (*p* < 0.001–0.003; ES = 1.78 to −11.86, large) in the EG. Otherwise, in the case of the CG, the only improvements were found in SPR10, RSAtotal, ISOQUAnd, ISOABDd, and ISOADDnd (*p* < 0.018; ES = −1.79 to −0.92, large). Furthermore, while in the EG a significant reduction in Q–H imbalance between baseline and post-training (*p* < 0.001; ES = 2.37–3.71, large) was observed, in the CG there was no significant change (*p* = 0.839–0.901; ES = –0.04 to 0.07, small). Finally, while in the CG the Abd–Add imbalance did not change between baseline and post-training (*p* = 0.142–0.340; ES = 0.32–0.51, small to moderate), in the EG there was a significant increase (*p* < 0.001; ES = –3.16 to –3.27, large).

Differences in the RPE_L_ and Hooper values between injured and non-injured for all players in the CG and EG are shown in Table 5. Significant differences (*p* < 0.01, ES = 0.74–2.14, moderate to large) were found in the RPE_L_ and the Hooper index between injured (*n* = 6) and non-injured (*n* = 14) players. For the EG, all RPE_L_ and Hooper values were higher (ES > 1.10, large) in injured compared to non-injured players. For the CG, no significant differences were found in the RPE_L_ or the Hooper index between injured (*n* = 5) and non-injured (*n* = 5) players for any period (*p* > 0.05). With regard to the CG, significant differences were found in the W1–12 period for the Hooper sleep values between injured and non-injured players (*p* = 0.029; ES = 1.68, large). In terms of Hooper’s stress values, significant differences were found between injured and non-injured players in the CG for the W1–6, W7–12, and W1–12 periods (*p* = 0.040–0.041; ES = 1.54–1.55, large). Significant differences were also found for Hooper’s fatigue values in the CG between injured and non-injured players in the W7–12 period (*p* = 0.049; ES = 1.46, large). Otherwise, there was no significant difference (*p* > 0.05) in the percentage change in the physical fitness tests between injured and non-injured players for the EG and CG.

## 4. Discussion

Approaching the study of sports injuries in soccer from a multifactorial approach can be very interesting for physical trainers to optimize player performance [63] and to try to prevent injuries [10,17,18,19,20,21,22]. Therefore, this study aimed: (1) to analyze the effects of a 12-week maximal strength-training program on injury incidence, injury burden, and physical fitness variables in semi-professional soccer players; and (2) to compare the RPE_L_ and state of well-being between injured and non-injured players. The main results showed that EG players experienced fewer injuries, less time of absence, and a lower injury burden than CG players, although it is quite likely that ‘luck’ played a role and that the nature of the injury influenced the documented burden. In addition, the EG had higher improvements in vertical jump, CODA, sprint, RSA, and isometric strength in comparison to the CG. All of these improvements were observed without increasing the load and without negatively influencing the state of well-being between the two groups. Finally, a reduction of the Q–H imbalance was observed in the EG after strength training, whereas no significant change in this variable was observed in the CG. The results obtained seem to show that maximal strength training has been effective at reducing injuries and improving the physical fitness of the players.

The collection, refinement, analysis, interpretation, and dissemination of loading data are usually performed with the aim of improving player performance and/or managing injury risk [64]. To achieve these results, soccer practitioners try to optimize the load at different stages of the training process, through several strategies such as the adjustment of individual sessions, the day-to-day planning, the periodization of the season, and the management of players with a long-term vision [64]. However, other authors indicate that the training or match load can show whether the planned load was realized by the player but in no case can it predict when a player will be injured [19]. Moreover, not controlling training and competition loads adequately can be a trigger for cumulative tissue overload and expose players to potentially injurious situations [33]. In this study, quantification of the training and match load was performed on the players allocated to the CG and EG, which is not common practice in studies analyzing the effect of a strength-training program [32]. The main results showed non-significant differences in the RPE_L_ and Hooper status between groups (i.e., CG vs. EG) overall in periods W1–6, W7–12, and W1–12, although there were some differences in spot weeks. Although the EG players performed additional maximum strength training sessions twice a week, the RPE_L_ and Hooper index of well-being were similar to those in the CG. In semi-professional soccer players, there is usually less control of these parameters due to limited resources and players must also juggle other occupational activities [15,16]. Thus, carrying out a maximal strength-training program in addition to regular soccer training did not produce a greater load for the players or negatively influence their well-being state, which is a positive aspect of this program. However, it would be interesting to be able to carry out load control in a differentiated manner between specific soccer training sessions and strength training sessions in future studies in order to know whether this load regulation is carried out through compensation in the soccer training sessions because it would be an aspect that could be affected. Interestingly, in the W1–12 and W1–6 periods, a higher Hooper fatigue value was observed in the EG with respect to the CG but this difference disappeared in the W7–12 period. At the beginning of the training protocol, the players possibly perceived greater fatigue, which due to the adaptation to the training loads was subsequently mitigated in the final phase of the intervention period. In this sense, these data show that adequate planning of the loads in the training sessions could be both important and necessary.

There is limited knowledge on analyzing the effectiveness of a maximal strength-training program for injury incidence in soccer. Making players available to compete is key to optimizing the team’s sporting performance [8] and for the club’s economic interests [6]. In our study, although no significant differences in injury incidence were observed between the EG compared to the CG, players in the CG suffered five injuries whereas those in the EG suffered only one. On the other hand, a lower injury burden was recorded for the EG compared to the CG. It may be striking that significant differences were found in burden but not in injury incidence, an aspect that some authors attribute to both the duration of the training program and the number of players involved in the research [20]. Therefore, the increase in burden may be an important indicator of injury severity, even when injury incidence does not vary significantly [4,14]. The lower injury burden (days of absence/1000 h of exposure) observed in the EG compared to the CG may be because maximal strength training improves coordination between motor units and the muscle innervated by them [65], and increases the length of the muscle fascicle [66,67]. Likewise, it has been shown that a longer muscle fiber can probably generate a greater maximal force [68]. Furthermore, other authors observed that for every 0.5 cm increase in fascicle length the risk of injury is reduced by 73.9% [21]. In addition, three of the four strength exercises performed in this study (i.e., unilateral horizontal leg press, unilateral lateral leg press with 45° inclination of the supporting foot with respect to the surface and knee flexion) have been largely focused on the work and activation of the gluteus medius and hamstrings. In recent years, the importance of lumbo-pelvic muscle function in the prevention and treatment of hamstring injuries has been proven, demonstrating how gluteus medius activation and strength during running has a high relationship with hamstring injuries [26]. Furthermore, it has been revealed that the greatest leg asymmetry (LA) between the right and left lower limbs occurs in the gluteal muscles and the least in the quadriceps [27]. The baseline versus post-training or change in LA between pre- and post-training was similar in the EG and CG, so it seems that this was not a determining factor for the difference between the number of injuries and injury burden in players who participated in this study. However, in the present study, in the CG the Q–H imbalance was maintained between pre- and post-training, whereas the EG significantly reduced the Q–H imbalance in both the right and left leg. This may have been an important factor in reducing the number of injuries in the EG compared to the CG. These results appear to be consistent with previous studies that have shown that improving the Q–H imbalance may be a positive factor in preventing lower-limb musculoskeletal injuries and improving decompensations in the pelvis [17,69], which could cause lateral pelvic tilts and increase pubic symphysis shear forces [69]. This could be favorable for reducing the number of injuries and burden. However, while in the CG the Abd–Add imbalance did not change, curiously in the EG there was a significant increase after the training protocol. However, it appears that the increase in Abd–Add imbalance did not negatively affect the number of injuries and burden. Possibly, as observed in previous studies, strength training that is focused on reducing the Q–H imbalance is more relevant or beneficial than improving the Abd–Add imbalance [17,69], although more studies are necessary to analyze the influence of the Abd-Add imbalance in lower extremity injuries. 

This maximal strength-based training program also produced improvements in most of the physical condition tests performed (i.e., vertical jump performance, 505-CODA, linear sprint test, RSA, and isometric strength) in the players who undertook it compared to their counterparts. These results support those shown in prior studies after the application of different maximal strength-training programs [28,29,70]. Specifically, Bogdanis et al. [28] showed that maximal strength training (i.e., 90% of 1 RM) had better results on maximal strength and RSA with respect to a moderate load training (i.e., 70% of 1 RM) in professional soccer players [28]. Other authors examined the physical capacity of a high-level elite soccer team during their preseason by applying a training program of maximum loads using 4 repetitions × 4 sets simultaneously twice a week for 8 weeks, observing improvements in maximal strength (51.7%), 10 m sprint (3.2%), and CMJ (5.2%) [29]. In these two studies, it was observed that increased strength does not produce changes in muscle mass, supporting the theory of neural adaptation and improved muscle strength without increased body mass, which is advantageous and undesirable for this sport [9,29]. It is possible that the pennation angle is of importance because it has been shown that for two muscles of the same volume, the muscle with fibers that attach at greater angles to the tendon will have the greater cross-sectional area and therefore greater capacity for maximal force production [71]. Furthermore, with this type of training, tendons with greater stiffness and thickness are achieved, increasing the rate of power and maximal force [72]. On the other hand, strength training aims to develop physical aspects that determine CODA speed, which includes muscular strength and power [73,74]. Some authors clarified that the different CODA actions were either strength- or speed-oriented, depending on the speed of approach and the angle of change in direction. It has been suggested that modest CODA angles of <90° are more speed-oriented, whereas angles of >90° are more force-oriented [75]. Therefore, maximal strength training may be more beneficial for improving strength-oriented (>90°) CODA, as is the case in our study. Furthermore, it is possible that specific maximum strength training could be beneficial for improving the performance of soccer players in terms of physical fitness. Consequently, considering that the EG improves most of the physical capacities analyzed to a greater extent and that it suffers a reduction in the number of injuries and burden, maximum strength training with characteristics similar to those used in the present study could be beneficial for semi-professional soccer players.

In the present study, it has been observed that for all the soccer players, significant differences (*p* < 0.01) were found in RPE_L_ and Hooper’s state of well-being between injured (*n* = 6) and non-injured (*n* = 14) players. Non-injured players showed a greater load and higher Hopper values than injured players. Despite these differences in RPE_L_ values, the results do not reflect that a greater load in training and competition is responsible for suffering fewer injuries. The differences found may simply be that the injured players have spent less time training due to being injured and have spent more time absent or unavailable, an aspect that may have conditioned the training load. Regarding Hooper’s well-being index, although previous studies have shown that injured players may have high levels of stress and anxiety due to not competing regularly [5], in our study the values obtained by injured players are lower than those of non-injured players. This aspect shows that players who completed a greater number of training sessions and matches may have an increase in sleep, fatigue, muscle soreness, and stress compared to players who were injured and completed a lower number of training sessions and matches. On the other hand, in this study no significant differences (*p* > 0.05) were observed in the percentage change in physical fitness tests between injured and non-injured players or between the EG and CG. The EG players suffered fewer injuries and had greater improvements and changes in physical fitness tests, although it is possible that a greater change in physical fitness did not directly influence injury reduction. An optimal improvement in physical fitness may be necessary, especially associated with some specific capabilities. Therefore, it would be interesting to carry out more studies that included a larger sample of players.

This study is not without limitations, the first being that it is only one team, so its special characteristics could influence the results. Thus, it would be interesting for future studies to apply this training to a larger number of teams at different competitive levels. Second, the results should be considered with caution if this training program were to be included in young players because the program was applied in senior players with very specific demands and age can be a decisive factor in the volume and intensity of training. Third, a program of longer duration could explain whether the effectiveness of the program also has a positive impact on the reduction of injury incidence because in our study we only found significant differences in the burden. Therefore, it would be interesting in future studies to: (1) quantify injury incidence and injury burden over a longer time interval (the whole season); (2) study how environmental factors affect injury incidence and injury burden; and (3) know the effect of prescribing maximal strength training in different periods of the season.

## 5. Conclusions

Performing a maximum strength-training program in addition to regular soccer training reduces the injury burden (i.e., days of absence/1000 h of exposure) and improves performance in jumping ability, change in direction, sprinting, RSA, and isometric strength, without increasing the load on the players or negatively influencing their state of well-being. In addition, this type of training has shown improvement in Q–H imbalance, an interesting aspect that could intervene in better muscle balance and thereby influence a reduction in the number and burden of injuries. Finally, we invite you to implement this specific practical training application.

## Figures and Tables

**Figure 1 healthcare-11-03195-f001:**
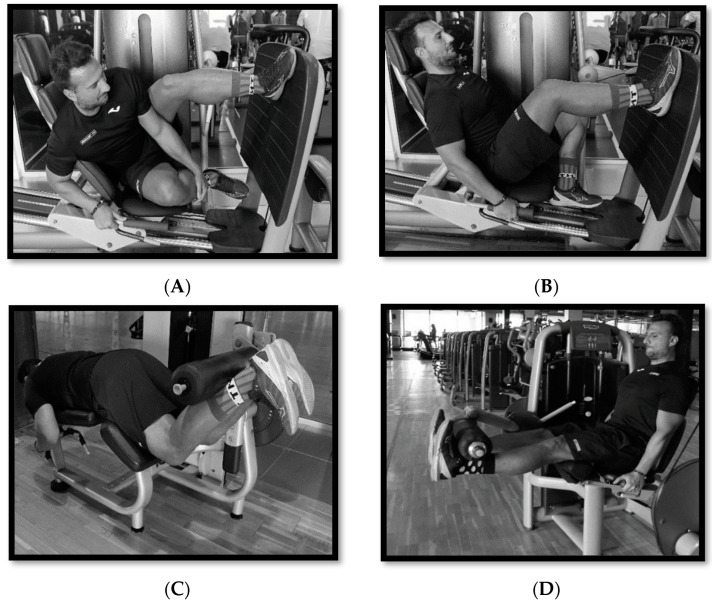
The selected exercises in the 12-week maximal strength-training program: horizontal leg press (**A**), unilateral lateral leg press with 45° inclination of the supporting foot with respect to the surface (**B**), knee extension (**C**), and knee flexion (**D**) in the 12-week maximal strength-training program.

**Figure 2 healthcare-11-03195-f002:**
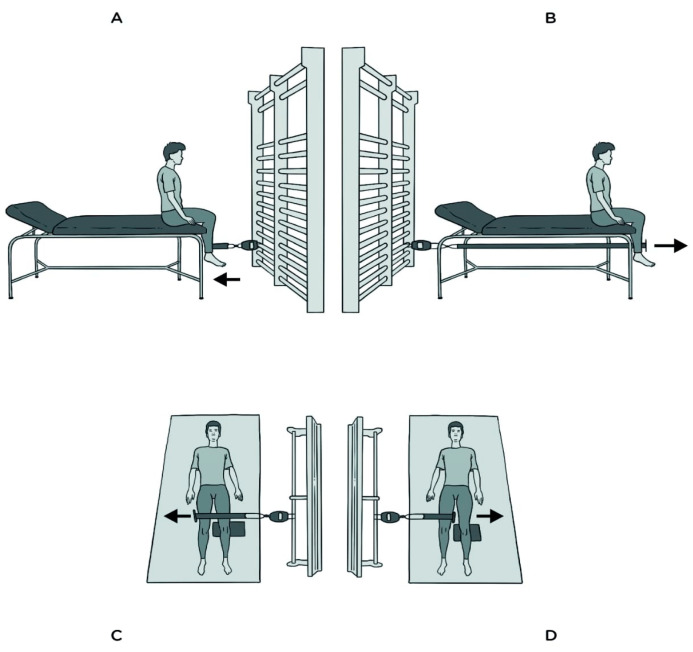
Isometric strength exercises for hamstrings (**A**), quadriceps (**B**), hip abductors (**C**), and hip adductors (**D**) muscles.

**Figure 3 healthcare-11-03195-f003:**
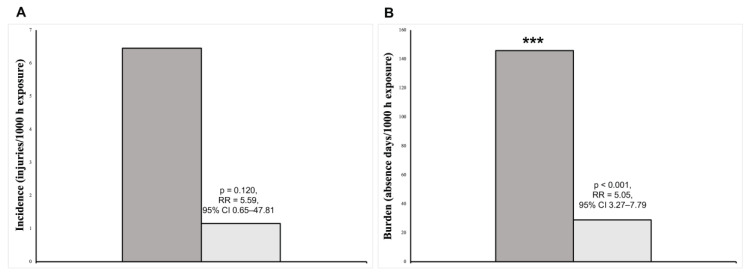
Between-groups differences in injury incidence (**A**) and burden (**B**). CG: control group; EG: experimental group; RR: rate ratio; CI: confidence interval. *** Significant differences (*p* < 0.001).

**Table 1 healthcare-11-03195-t001:** Types of exercises, volume, intensity, and recovery time for the 12-week intervention training period.

Exercise	Temporal Sequence of Training Program	Series	Repetitions	Rest between Sets
Unilateral horizontal leg Press	W1–2: 85% 1 RM W3–4: 90% 1 RM W5–12: 95% 1 RM	3	W1–2: 4 Rep. W3–4: 4 Rep. W5–12: 3 Rep.	3 min
Lateral leg press with 45° support	W1–2: 85% 1 RM W3–4: 90% 1 RM W5–12: 95% 1 RM	3	W1–2: 4 Rep. W3–4: 4 Rep. W5–12: 3 Rep.	3 min
Knee extension	W1–2: 85% 1 RM W3–4: 90% 1 RM W5–12: 95% 1 RM	3	W1–2: 4 Rep. W3–4: 4 Rep. W5–12: 3 Rep.	3 min
Knee flexion	W1–2: 85% 1 RM W3–4: 90% 1 RM W5–12: 95% 1 RM	3	W1–2: 4 Rep. W3–4: 4 Rep. W5–12: 3 Rep.	3 min

Note. W: week; Rep.: repetitions; 1 RM: 1 repetition maximum; min: minutes.

**Table 2 healthcare-11-03195-t002:** Comparison between the RPE_L_ and Hooper values between players performing experimental training (EG, n = 10) and the control group (CG, n = 10).

	RPE_L_ (AU)					Hooper Total (AU)				
	CG	EG	%Diff	*p*	ES	CG	EG	%Diff	*p*	ES
W1	3032.5 ± 417.9	3144.0 ± 415.2	3.55	0.557	−0.27	61.3 ± 10.6	64.9 ± 11.2	5.55	0.468	−0.33
W2	2952.0 ± 427.7	2978.5 ± 927.7	0.89	0.936	−0.04	58.8 ± 9.2	57.6 ± 19.4	−2.08	0.862	0.08
W3	2703.3 ± 538.3	3362.8 ± 361.0	19.61	0.008 **	−1.44	53.4 ± 22.7	59.9 ± 23.0	10.85	0.533	−0.29
W4	3015.0 ± 406.2	3310.0 ± 299.3	8.91	0.095	−0.84	47.9 ± 26.4	64.4 ± 10.4	25.62	0.082	−0.82
W5	2647.2 ±1025.3	3247.5 ± 465.7	18.48	0.113	−0.77	51.7 ± 26.0	64.6 ± 8.0	19.97	0.150	−0.67
W6	3010.0 ± 497.8	3216.7 ± 291.5	6.42	0.298	−0.51	54.9 ± 21.8	62.1 ± 17.7	11.59	0.428	−0.36
W7	2851.1 ± 383.2	3256.1 ± 318.1	12.44	0.027 *	−1.15	55.3 ± 20.9	57 ± 21.3	2.98	0.859	−0.08
W8	2744.3 ±1083.0	3017.7 ± 416.2	9.06	0.490	−0.33	51.6 ± 27.8	54.7 ± 20.1	5.67	0.778	−0.13
W9	2583.9 ±1064.1	2989.0 ± 337.0	13.55	0.268	−0.53	49.4 ± 27.7	57.6 ± 4.0	14.24	0.366	−0.42
W10	2671.5 ± 804.0	3061.0 ± 360.7	12.72	0.188	−0.65	45.3 ± 23.5	69.7 ± 9.8	35.01	0.007 **	−1.36
W11	2585.4 ± 926.9	2999.4 ± 410.0	13.80	0.213	−0.58	58.3 ± 23.3	60.1 ± 12.1	3.00	0.831	−0.10
W12	2621.0 ± 431.3	3214.0 ± 317.1	18.45	0.003 **	−1.57	57.5 ± 18.7	63.1 ± 8.0	8.87	0.394	−0.39
W1–6	15,921.0 ± 4142.2	18,601.5 ± 2523.4	14.41	0.098	−0.78	328 ± 89.2	373.5 ± 64.3	12.18	0.207	−0.59
W7–12	14,705.0 ± 5200.0	17,909.8 ± 2504.6	17.89	0.096	−0.79	317.4 ± 115.1	362.2 ± 55.8	12.37	0.283	−0.50
W1–12	30,626.0 ± 7966.5	36,511.3 ± 4302.4	16.12	0.055	−0.92	645.4 ± 174.0	735.7 ± 114.3	12.27	0.187	−0.61

Note. W: week; RPE_L_: perceived exertion load; %Diff: difference in percentage between groups; *p* = level of significance; ES: effect size. * *p* < 0.05; ** *p* < 0.01 significant differences between groups.

**Table 3 healthcare-11-03195-t003:** The number of injuries, mechanism, type, region, time of injury, and absence time values of the experimental (EG, n = 10) and control group (CG, n = 10) players.

	Total	CG	EG
**Injuries (n)**	6 (100%)	5(83.3%)	1(16.7%)
**Type of injury**			
Muscular	3 (50%)	3 (50%)	0
Ligament	2 (33.3%)	1 (16.7%)	1 (16.7%)
Tendon	1 (100%	1 (100%)	0
**Mechanism of injury**			
Direct	0	0	0
Indirect	6 (100%)	5(83.3%)	1(16.7%)
Overuse	0	0	0
**Body region**			
Thigh	3 (50%)	3 (50%)	0
Knee	2 (33.3%)	2 (33.3%)	0
Ankle	1 (16.7%)	0	1 (16.7%)
**Musculoskeletal structure**			
Hamstrings	3 (50%)	3 (50%)	0
Knee ligament	1 (16.7%)	1 (16.7%)	0
Patellar tendon	1 (16.7%)	1 (16.7%)	0
Ankle ligament	1 (16.7%)	0	1 (16.7%)
**Time and epoch**			
Training session	3 (50%)	2 (33.3%)	1 (16.7%)
Competition session	3 (50%)	3 (50%)	0
W1–6	4 (66.6%)	3 (50%)	1 (16.7%)
W7–12	2 (33.3%)	2 (33.3%)	0
**Time of absence (days)**			
Total	138 (100%)	113 (81.9%)	25(18.1%)
Mean ± SD	34.5 ± 16.6	37.7 ± 18.8	

Note. %: the percentage values correspond to the percentage with respect to the total number of injuries.

**Table 4 healthcare-11-03195-t004:** Changes in physical fitness attributes after the 12-week period intervention of the control group (CG, n = 10) and experimental group (EG, n = 10).

	CG					EG					Between-Group Differences	
Variables	Baseline	Post	%Diff	*p*	ES	Baseline	Post	%Diff	*p*	ES	*p*	F
CMJ (cm)	37.93 ± 3.76	38.02 ± 3.90	0.24	0.468	−0.24	36.69 ± 4.58	39.98 ± 4.63	8.23	<0.001 ***	−7.51	<0.001 ***	294.93
CMJd (cm)	22.27 ± 2.79	22.20 ± 2.88	−0.32	0.572	0.19	19.79 ± 4.19	21.36 ± 4.03	7.35	<0.001 ***	−5.26	<0.001 ***	95.19
CMJnd (cm)	22.73 ± 3.50	22.78 ± 3.41	0.22	0.630	−0.16	21.20 ± 4.13	22.67 ± 4.22	6.48	<0.001 ***	−3.60	<0.001 ***	67.90
CMJ_LA_ (%)	−1.85 ± 5.75	−0.28 ± 1.41	−556.38	0.452	−0.25	−7.85 ± 9.21	−7.09 ± 1.82	−10.69	0.812	−0.08	<0.001 ***	81.48
SJ (cm)	31.04 ± 3.97	30.97 ± 3.86	−0.23	0.563	0.19	29.52 ± 3.61	31.48 ± 3.35	6.23	<0.001 ***	−3.96	<0.001 ***	114.50
EI (%)	23.55 ± 15.55	24.06 ± 15.85	−0.25	0.452	0.09	24.59 ± 9.85	27.16 ± 8.91	−0.93	0.016 *	−0.30	0.075	3.58
505-CODAd (s)	2.26 ± 0.07	2.26 ± 0.09	−0.09	0.785	0.09	2.31 ± 0.06	2.21 ± 0.05	−4.80	<0.001 ***	5.77	<0.001 ***	108.68
505-CODnd (s)	2.26 ± 0.08	2.26 ± 0.07	−0.09	0.764	0.10	2.29 ± 0.07	2.18 ± 0.06	−4.99	<0.001 ***	3.83	<0.001 ***	115.22
505-CODA_LA_ (%)	−0.18 ± 1.80	−0.23 ± 2.48	19.91	0.931	0.03	0.95 ± 1.89	1.12 ± 1.68	15.76	0.715	−0.12	0.555	0.36
SPR10 (s)	1.70 ± 0.07	1.69 ± 0.06	−0.77	0.018 *	0.92	1.74 ± 0.07	1.65 ± 0.07	−5.21	<0.001 ***	1.94	<0.001 ***	20.10
SPR20 (s)	2.93 ± 0,10	2.91 ± 0.10	−0.55	0.168	0.47	2.99 ± 0.11	2.87 ± 0.13	−4.11	<0.001 ***	1.64	0.002 **	13.74
SPR40 (s)	5.31 ± 0.16	5.29 ± 0.15	−0.30	0.112	0.56	5.40 ± 0.19	5.25 ± 0.20	−2.84	<0.001 ***	1.64	<0.001 ***	16.23
RSAtotal (s)	21.85 ± 0.43	21.71 ± 0.45	−0.64	0.010 *	1.02	22.34 ± 0.70	21.50 ± 0.66	−3.92	<0.001 ***	5.34	<0.001 ***	87.34
RSASdec (s)	3.11 ± 0.95	3.04 ± 0.68	−2.33	0.775	0.09	4.15 ± 1.24	3.10 ± 0.76	−33.94	0.003 **	1.25	0.141	2.38
ISOQUAd (kg)	40.87 ± 5.19	41.09 ± 5.04	0.54	0.073	−0.64	40.30 ± 4.06	44.24 ± 3.84	8.91	<0.001 ***	−10.96	<0.001 ***	741.72
ISOQUAnd (kg)	39.20 ± 4.77	39.51 ± 4.76	0.78	0.003 **	−1.28	39.25 ± 4.48	43.29 ± 4.48	9.33	<0.001 ***	−11.86	<0.001 ***	753.38
ISOQUA_LA_ (%)	3.77 ± 7.92	3.59 ± 7.49	−5.10	0.462	0.24	2.45 ± 8.55	2.04 ± 7.91	−19.84	0.374	0.30	0.471	0.54
ISOHAMSd (kg)	22.57 ± 3.05	22.68 ± 2.91	0.49	0.281	−0.36	22.01 ± 2.55	26.22 ± 2.58	16.06	<0.001 ***	−9.96	<0.001 ***	608.11
ISOHAMSnd (kg)	22.33 ± 2.82	22.55 ± 3.01	0.98	0.051	−0.71	22.00 ± 2.93	26.27 ± 3.04	16.25	<0.001 ***	−6.97	<0.001 ***	349.22
ISOHAMS_LA_ (%)	0.83 ± 6.23	0.51 ± 6.14	−61.76	0.648	0.15	0.06 ± 7.16	−0.21 ± 7.15	130.19	0.711	0.12	0.994	5.93
Imb.Q-H Right (%)	44.79 ± 2.33	44.81 ± 2.15	0.04	0.901	−0.04	45.41 ± 2.60	40.75 ± 2.18	−11.43	<0.001 ***	3.71	<0.001 ***	139.39
Imb.Q-H Left (%)	43.05 ± 1.94	42.99 ± 2.19	−0.13	0.839	0.07	44.00 ± 3.23	39.36 ± 2.08	−11.78	<0.001 ***	2.37	<0.001 ***	56.70
ISOABDd (kg)	29.81 ± 5.59	30.16 ± 5.58	1.16	<0.001 ***	−1.79	27.93 ± 4.65	32.36 ± 4.66	13.69	<0.001 ***	−8.00	<0.001 ***	438.81
ISOABDnd (kg)	28.75 ± 4.54	28.83 ± 4.45	0.28	0.380	−0.29	27.87 ± 4.24	31.51 ± 4.02	11.55	<0.001 ***	−9.22	<0.001 ***	651.64
ISOABD_LA_ (%)	2.86 ± 7.38	3.69 ± 7.75	22.47	0.011 *	−1.02	−0.34 ± 9.46	2.12 ± 8.58	116.25	0.009 **	−1.05	0.086	3.32
ISOADDd (kg)	24.61 ± 4.00	25.89 ± 4.78	4.94	0.234	−0.40	23.97 ± 2.91	24.57 ± 2.79	2.44	<0.001 ***	−1.78	0.489	0.50
ISOADDnd (kg)	24.41 ± 3.65	24.67 ± 3.74	1.05	0.011 *	−1.00	24.18 ± 2.89	24.62 ± 2.84	1.79	<0.001 ***	−1.86	0.130	2.53
ISOADD_LA_ (%)	0.51 ± 5.91	3.48 ± 11.98	85.25	0.338	−0.32	−1.02 ± 5.61	−0.29 ± 5.08	−254.36	0.201	−0.44	0.510	0.45
Imb.Abd-Add Right (%)	16.83 ± 6.51	13.76 ± 8.33	−22.29	0.340	0.32	13.19 ± 9.46	23.53 ± 6.67	43.94	<0.001 ***	−3.27	<0.001 ***	16.45
Imb.Abd-Add Left (%)	14.76 ± 6.09	14.11 ± 6.76	−4.67	0.142	0.51	12.56 ± 7.53	21.56 ± 5.46	41.76	<0.001 ***	−3.16	<0.001 ***	103.93

Note. CMJ: counter movement jump; CMJd: dominant leg counter movement jump; CMJnd: non-dominant leg counter movement jump; CMJ_LA_: leg asymmetry in counter movement jump; SJ: squat jump; 505-CODAd: dominant leg change in direction ability; 505-CODAnd: non-dominant leg change in direction ability; 505-CODA_LA_: leg asymmetry in change in direction ability; SPR10: linear sprint in 10 m; SPR20: linear sprint in 20 m; SPR40: linear sprint in 40 m; RSAtotal: repeated sprint ability total; RSASdec: repeated sprint ability fatigue index; ISOQUAd: dominant leg isometric strength in quadriceps muscles; ISOQUAnd: non-dominant leg isometric strength in quadriceps muscles; ISOQUA_LA_: leg asymmetry in quadriceps; ISOHAMSd: isometric strength in hamstrings muscles in dominant leg; ISOHAMSnd: isometric strength in hamstrings muscles in non-dominant leg; ISOHAMS_LA_: leg asymmetry in hamstrings; Imb.Q-H Right: imbalance quadriceps–hamstrings in right leg; Imb.Q-H Left: imbalance quadriceps–hamstrings in left leg; ISOABDd: isometric strength in abductors muscles in dominant leg; ISOABDnd: isometric strength in abductors muscles in non-dominant leg; ISOABD_LA_: leg asymmetry in abductors; ISOADDd: isometric strength in adductors muscles in dominant leg; ISOADDnd: isometric strength in adductors muscles in non-dominant leg; ISOADD_LA_: leg asymmetry in adductors; Imb.Abd-Add Right: imbalance abductors–adductors in right leg; Imb.Abd-Add Left: imbalance abductors–adductors in left leg; %Diff: difference in percentage between groups; *p* = level of significance; ES: Effect size. * *p* < 0.05, ** *p* < 0.01 *** *p* < 0.001: significant differences between baseline and post or between groups.

**Table 5 healthcare-11-03195-t005:** Comparison between the RPE_L_ and Hooper values between the injured and not-injured for all players, the control group (CG, n = 10), and the experimental group (EG, n = 10).

	CG				EG				All			
Variables	Injured (n = 6)	Not Injured (n = 4)	%Diff	p; ES	Injured (n = 1)	Not Injured (n = 9)	%Diff	ES	Injured (n = 7)	Not Injured (n = 13)	%Diff	p; ES
**RPE_L_**												
W1–6	13,763.0 ± 5153.0	18,079.0 ± 639.3	−31.36	0.134; 1.18	15,990.0	18,891.7 ± 2493.3	−18.15	1.16	14,134.2 ± 4697.8	18,601.4 ± 2028.5	−31.61	0.009 **; 0.74
W7–12	12,242.0 ± 6415.7	17,168.0 ± 2123.9	−40.24	0.166; 1.03	10,931.0	18,685.2 ± 541.1	−70.94	14.33	12,023.5 ± 5763.3	18,143.4 ± 1462.0	−50.90	0.006 **; 0.76
W1–12	26,005.0 ± 9263.6	35,247.0 ± 1897.0	−35.54	0.089; 1.38	26,921.0	37,576.9 ± 2837.3	−39.58	3.76	26,157.7 ± 8294.1	36,744.8 ± 2720.9	−40.47	<0.001 ***; 2.14
**Hooper total**												
W1–6	279.0 ± 101.6	377.0 ± 39.6	−35.13	0.099; 1.27	293.0	382.4 ± 61.3	−30.53	1.46	281.3 ± 91.1	380.5 ± 53.0	−35.25	0.006 **; 1.51
W7–12	253.6 ± 133.8	381.2 ± 41.8	−50.32	0.100; 1.29	231.0	376.8 ± 33.3	−63.11	4.38	249.8 ± 120.1	378.4 ± 35.0	−51.44	0.001 **; 1.84
W1–12	532.6 ± 183.8	758.2 ± 50.3	−42.36	0.050; 1.67	524.0	759.2 ± 92.1	−44.90	2.55	531.2 ± 164.5	758.9 ± 77.4	−42.87	<0.001 ***; 2.10
**Hooper Sleep**												
W1–6	44.40 ± 18.53	66.20 ± 10.3	−49.10	0.051; 1.45	51.0	67.6 ± 9.6	−32.55	1.73	45.5 ± 16.8	67.1 ± 9.5	−47.47	0.002 **; 1.80
W7–12	39.6 ± 24.8	64.0 ± 13.0	−61.62	0.087; 1.23	43.0	62.3 ± 8.8	−44.88	2.20	40.2 ± 22.2	62.9 ± 10.0	−56.47	0.054; 1.32
W1–12	84.0 ± 35.1	130.2 ± 16.9	−55.00	0.029 *; 1.68	94.0	129.9 ± 17.0	−66.48	2.11	85.7 ± 31.7	130.0 ± 16.3	−51.69	<0.001 ***; 2.04
**Hooper Stress**												
W1–6	53.2 ± 16.3	81.4 ± 20.2	−53.01	0.041 *; 1.54	43.0	68.0 ± 12.2	−58.14	2.04	51.5 ± 15.1	72.8 ± 16.2	−41.36	0.013 *; 1.34
W7–12	43.0 ± 18.8	66.0 ± 9.4	−53.49	0.040 *; 1.55	48.0	66.6 ± 10.7	−38.75	1.74	43.8 ± 16.9	66.4 ± 9.9	−51.60	0.001 **; 1.84
W1–12	105.2 ± 41.0	164.6 ± 36.1	−56.46	0.041 *; 1.54	87.0	135.3 ± 20.6	−55.52	2.35	102.2 ± 37.4	145.8 ± 29.5	−42.66	0.012 *; 1.37
**Hooper Fatigue**												
W1–6	82.8 ± 32.8	107.4 ± 10.7	−29.71	0.173; 1.01	95.0	121.0 ± 21.8	−27.37	1.19	84.8 ± 29.7	116.1 ± 19.3	−36.91	0.011 *; 1.38
W7–12	53.2 ± 17.5	80.8 ± 20.2	−51.88	0.049 *; 1.46	40.0	68.2 ± 12.4	−70.50	2.28	51.0 ± 16.5	72.7 ± 16.1	−42.55	0.013 *; 1.34
W1–12	160.4 ± 54.0	217.4 ± 8.8	−35.54	0.077; 1.48	167.0	239.8 ± 32.4	−43.59	2.25	161.5 ± 48.3	231.8 ± 28.2	−43.53	<0.001 ***; 2.01
**Hooper Muscle soreness**												
W1–6	98.6 ± 36.1	122.0 ± 7.9	−23.73	0.224; 0.90	104.0	125.9 ± 20.0	−21.06	1.10	99.5 ± 32.4	124.5 ± 16.4	−25.13	0.122; 0.97
W7–12	82.8 ± 33.4	107.0 ± 9.4	−29.23	0.184; 0.99	92.0	120.0 ± 21.9	−30.43	1.28	84.3 ± 30.1	115.4 ± 19.1	−36.89	0.012 *; 1.37
W1–12	183.0 ± 60.6	246.0 ± 13.9	−34.43	0.080; 1.43	176.0	254.2 ± 26.8	−44.43	2.92	181.8 ± 54.3	251.3 ± 22.8	−38.23	0.024 *; 1.67

Note. CG: control group; EG: experimental group; W: week; All: all groups; %Diff: difference in percentage between groups; *p* = level of significance; ES: Effect size. * *p* < 0.05, ** *p* < 0.01, *** *p* < 0.001: significant differences between baseline and post or between groups.

## Data Availability

Data are unavailable due to the privacy of participants.

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
