# Peer review of "Is a Maximal Strength-Training Program Effective on Physical Fitness, Injury Incidence, and Injury Burden in Semi-Professional Soccer Players? A Randomized Controlled Trial"

_healthcare, 2023, doi:10.3390/healthcare11243195_

Round 1

Reviewer 1 Report

Comments and Suggestions for Authors

Dear authors,

Your study deals with a topic that has been the subject of much research for many years. In particular, the 11+ program imposed by FIFA, which is not purely strength training but includes strengthening exercises, can claim to be effective [1,2]. However, there are also studies that clearly show that strength training in addition to soccer training improves performance [3,4]. The strength of your work probably lies in the fact that you cover several sub-areas: performance, load, injuries. These three aspects are rarely surveyed together in such abundance in one study and give the reader a comprehensive picture.

Unfortunately, the insufficient number of participants in this study in a parallel group design is a massive problem. The results are not statistically reliable. You would have to explain to me how you arrived at the calculated sample size? I have used G*Power to determine the necessary number of cases myself and always come up with larger values. But I would be happy to be instructed!

I have inserted my comments directly into the PDF. If you have any questions, please do not hesitate to ask.

Best regards

[1] Gomes Neto M, Conceição CS, de Lima Brasileiro AJA, de Sousa CS, Carvalho VO, de Jesus FLA. Effects of the FIFA 11 training program on injury prevention and performance in football players: a systematic review and meta-analysis. Clin Rehabil. 2017;31(5):651-659. doi:10.1177/0269215516675906

[2] Al Attar WSA, Alshehri MA. A metaanalysis of metaanalyses of the effectiveness of FIFA injury prevention programs in soccer. Scand J Med Sci Sports. 2019;00:1–10. https ://doi.org/10.1111/sms.13535

[3] Pardos-Mainer E, Lozano D, Torrontegui-Duarte M, Cartón-Llorente A, Roso-Moliner A. Effects of Strength vs. Plyometric Training Programs on Vertical Jumping, Linear Sprint and Change of Direction Speed Performance in Female Soccer Players: A Systematic Review and Meta-Analysis. Int J Environ Res Public Health. 2021;18(2):401. Published 2021 Jan 6. doi:10.3390/ijerph18020401

[4] Sánchez M, Sanchez-Sanchez J, Nakamura FY, Clemente FM, Romero-Moraleda B, Ramirez-Campillo R. Effects of Plyometric Jump Training in Female Soccer Player's Physical Fitness: A Systematic Review with Meta-Analysis. Int J Environ Res Public Health. 2020;17(23):8911. Published 2020 Nov 30. doi:10.3390/ijerph17238911

Author Response

Your study deals with a topic that has been the subject of much research for many years. In particular, the 11+ program imposed by FIFA, which is not purely strength training but includes strengthening exercises, can claim to be effective [1,2]. However, there are also studies that clearly show that strength training in addition to soccer training improves performance [3,4]. The strength of your work probably lies in the fact that you cover several sub-areas: performance, load, injuries. These three aspects are rarely surveyed together in such abundance in one study and give the reader a comprehensive picture.

[1] Gomes Neto M, Conceição CS, de Lima Brasileiro AJA, de Sousa CS, Carvalho VO, de Jesus FLA. Effects of the FIFA 11 training program on injury prevention and performance in football players: a systematic review and meta-analysis. Clin Rehabil. 2017;31(5):651-659. doi:10.1177/0269215516675906

[2] Al Attar WSA, Alshehri MA. A meta‐analysis of meta‐analyses of the effectiveness of FIFA injury prevention programs in soccer. Scand J Med Sci Sports. 2019;00:1–10. https ://doi.org/10.1111/sms.13535

[3] Pardos-Mainer E, Lozano D, Torrontegui-Duarte M, Cartón-Llorente A, Roso-Moliner A. Effects of Strength vs. Plyometric Training Programs on Vertical Jumping, Linear Sprint and Change of Direction Speed Performance in Female Soccer Players: A Systematic Review and Meta-Analysis. Int J Environ Res Public Health. 2021;18(2):401. Published 2021 Jan 6. doi:10.3390/ijerph18020401

[4] Sánchez M, Sanchez-Sanchez J, Nakamura FY, Clemente FM, Romero-Moraleda B, Ramirez-Campillo R. Effects of Plyometric Jump Training in Female Soccer Player's Physical Fitness: A Systematic Review with Meta-Analysis. Int J Environ Res Public Health. 2020;17(23):8911. Published 2020 Nov 30. doi:10.3390/ijerph17238911

Authors' Response (AR): We would like to express our gratitude for the time you have devoted to review our manuscript and provide us with useful comments and suggestions to improve the quality of our work. We have found your criticisms and recommendations to be constructive in all cases, addressing each revision point by point. Edits and changes have been highlighted in blue color within the text. We appreciate your positive assessment of our work. We agree that there are not too many works that approach this topic from different perspectives (performance, load and injuries).

Unfortunately, the insufficient number of participants in this study in a parallel group design is a massive problem. The results are not statistically reliable. You would have to explain to me how you arrived at the calculated sample size? I have used G*Power to determine the necessary number of cases myself and always come up with larger values. But I would be happy to be instructed!

AR: Thanks for your comments, we understand the reviewer's concern. However, it should be considered that in order to implement this type of study it is necessary that the soccer training and competition performed is similar in the control group (CG) and the experimental group (EG), so in most cases they are performed in a single team. This same limitation and therefore similar samples have been used in other relevant studies as indicated in L140 [41,42]. These studies used a similar design to ours, employing similar or smaller sample sizes. In addition, for this study the sample size of each group was calculated using G*Power software. Note that in G*Power it is necessary to enter the effect size value and select the type of study design/statistical analysis to be performed. It is possible that for this reason the data may not match. It should be borne in mind that these are longitudinal studies in which an intervention program is applied and we have a CG, with the difficulty that this entails.

[41] Brull-Muria, E.; Beltran-Garrido, J.V. Effects of a Specific Core Stability Program on the Sprint and Change-of-Direction Maneu-verability Performance in Youth, Male Soccer Players. Int J Environ Res Public Health 2021, 18, doi:10.3390/ijerph181910116.

[42] Hung, K.C.; Chung, H.W.; Yu, C.C.W.; Lai, H.C.; Sun, F.H. Effects of 8-Week Core Training on Core Endurance and Running Economy. PLoS One 2019, 14, doi:10.1371/journal.pone.0213158.

I have inserted my comments directly into the PDF. If you have any questions, please do not hesitate to ask.

AR: Thanks for the reviewers’ edits and observations. We have corrected all the issues and observations commented directly in the article. All your observations have been very helpful and have been very useful to us in improving the quality of our study, we greatly appreciate it. Below are all the changes that have been made throughout the manuscript:

1.The introduction should also look at the existing studies from the non-professional sector in order to show what has already been investigated, what effects training interventions have had and what new findings your own study will ultimately bring!

Here are possible sources with amateur players:

https://doi.org/10.3390/ijerph182413351

https://doi.org/10.1016%2Fj.jshs.2020.11.003

https://doi.org/10.1007/s40279-013-0026-0

https://doi.org/10.1123/ijspp.2020-0862

https://doi.org/10.1177/0363546515574057

https://doi.org/10.1177/0363546511419277

https://doi.org/10.1111/sms.13353

https://doi.org/10.1136/bjsports-2022-105890

https://doi.org/10.1123/jsr.2019-0390

https://doi.org/10.1136/bjsports-2012-091277

AR: Thank you for your comments. We have modified part of the introduction and added some of the references provided (L97-101). In addition, we have highlighted the novel contribution of our research. We are very grateful for this insightful commentary to improve our study.

  1. My a priori calculation with G*Power for an unpaired t-test (alpha = 0.05; power = 95%; Cohen's d = 0.8) results in a sample size of 84, i.e. 42 players per group! And for an ANCOVA still 28 participants are required.

AR: Thanks for sharing your concern. We have calculated the power statistic by a priori power analysis with a type I error rate of 0.05 (alpha = 5%) and 95% statistical power (beta) and effect size dz (0.8) for a related samples test. We hope this explanation help the reviewer to obtain the same result as authors.

  1. What is this assumption based on? Is there a reference to other studies?

AR: Thank you for the observation. Here, we include some studies to use similar statistical power calculations. In these papers 0.90 Cohen’s d is used.

10.1080/15438627.2021.2010205

10.1080/15438627.2022.2079989

10.1080/15438627.2021.2010206

  1. Which end of the scale indicates low stress?

AR: Thank you for this great observation. We have proceeded to make the appropriate changes to the text, changing "bad" to "low" and "good" to "high".

  1. As far as I know, the Hooper Index is the sum of the four items. And each item is rated from 1-7 or 1-10, whereby a lower value is better for all items! Why was the Sleep item measured using an opposite scale? Please check!

AR: Thanks for the comment. We have checked Hopper's index and it is indeed correct as stated in the text.

(48) Hooper, S.L.; Mackinnon, L.T. Monitoring Overtraining in Athletes. Recommendations. Sports Med 1995, 20, 321-7, doi: 10.2165/00007256-199520050-00003.

  1. The Shapiro-Wilk test should be used for small sample sizes! Please check the normal distribution using this test.

AR: Thank you very much for your observation. We apologize for the error, it is indeed a typing error, we have checked it and the test used to check the normality was Shapiro Wilk.

  1. Were the requirements met? If not, how was the analysis carried out?

AR: Thanks for the question. After assessing the normality of data and homogeneity of variances a paired t-test analysis was used.

  1. Effect sizes are always the amount of a result and are therefore positive. Please delete the minus sign for all effect sizes.

AR: We understand the reviewer concern. However, we consider indicating the positive o minus value attending to apply the formula to calculate the effect size. The sign indicates the direction of the effect size in favor to control or experimental group, or in favor to pre or post time measures.

  1. Add case numbers for the groups in the table header.

AR: Thanks for the observation. We have corrected the number of cases in all table headers.

  1. Add case numbers for the groups in the table header

The table is too wide; not all values can be viewed!

I find an analysis with even smaller groups very critical and would not include it in the paper! Especially the comparison of 1 injured player with 9 healthy players in the EG is not acceptable. Only the comparison of injured and non-injured players across all players would make sense. Regardless of this, the p-values for the EG would be missing in comparison to the other analyses in table 5.

AR: Thank you very much for your comment. We have added the number of players in each case both in the head and in the subgroups. Although the sample size in the GE injured and not injured is different (1 vs. 9), we think it is relevant to provide descriptive data as we consider this is one of the most novel parts of our study and we thank you very much for your appreciation.

Reviewer 2 Report

Comments and Suggestions for Authors

healthcare-2717649-peer-review-v1

Is a Maximal Strength Training Program Effective on Physical Fitness, Injury Incidence, and Injury Burden in Semi-Professional Soccer Players? A Randomized Controlled Trial

Journal: Healthcare

Thank you for the opportunity to review your manuscript. From reading your paper I understood your main findings to be that you demonstrated that a relatively heavy resistance training programme, in addition to traditional football training, reduces injury burden and improved fitness without unduly increasing load negatively affecting well-being. 

General comments 

The study seems to be very underpowered for outcomes like injury incidence and length of rehabilitation - would be interesting to see the inputs used in your power calculation.

While the writing and grammar is not poor, it is inconsistent - consider seeking assistance with English and proofreading.

e.g  L32 which players must be prepared to cope their optimal sport performance in each microcycle 

This does not make sense grammatically or otherwise. There are numerous occurrences like this throughout the manuscript. 

The introduction makes a strong case for conditioning as an injury prevention strategy without really giving coverage to the multiple factors thought to contribute to athletic injury, including ‘chance’, foul play, environmental factors, fatigue, warm up, style of play, and of course strength and conditioning. Although it was reassuring to see that training load and injury risk was reviewed. 

No detail is provided on the time of season that the intervention was undertaken. Was this preseason? What are the implications of trying to build strength during a competitive phase. Periodisation is mentioned but not explored relative to your study. 

I am assuming that the football specific training was 4 x 105 min sessions = 420 mins /week?

How did you find participants who had no previous neuromuscular injury?

The figures depicting the resistance training exercises were helpful - I think C is commonly identified as leg extension and D as a leg curl (the femur is not really moving!)

Descriptions of isometric testing are not clear on the joint angle tested and how these were standardised/normalised across participants.

Table 3 - musculoskeletal structure rather than muscular? Are internal and external necessary when referring to ligamentous injuries.

Figure 3 does not provide a scale or values 

Table 4 - I wonder about the value of this table - it could probably be summarised in text.

Discussion

Suggest care - you did not show that EG players experience fewer injuries statistically! There probably should be some commentary on the injury incidence in the EG group relative to norms for football - quite likely LUCK may have played a part and the nature of the injury likely had an influence on documented burden.

A very useful finding that the additional training load appeared to be well tolerated. 

The resistance exercise used were very afunctional, so it does seem a bit of stretch to suggest that muscle architecture and balance ratios are a reason for a lower burden.

I am not sure why the altered hip ABD/ADD balance was a surprise for the EG group given that you were targeting gluteus medius etc and ignoring the adductors. Given some arguments around groin injuries and symphysis pubis injury in football, I thought that this may have been explored further.  

Overall my view is that your discussion and conclusions are extending beyond what your data can support. In that respect I feel that the limitations need expansion and as noted the difficulty of isolating causation with injury needs to underpin discussions. The most significant finding for me was that extra RT did not adversely affect player wellbeing and perceived fatigue. 

Best wishes

Comments on the Quality of English Language

needs definite attention throughout - check grammar/sense

Author Response

Thank you for the opportunity to review your manuscript. From reading your paper I understood your main findings to be that you demonstrated that a relatively heavy resistance training programme, in addition to traditional football training, reduces injury burden and improved fitness without unduly increasing load negatively affecting well-being. 

Authors’ response (AR): We would like to express our gratitude for the time you have devoted in reviewing our manuscript and providing useful comments and suggestions to improve the quality of our work. We have found your criticisms and recommendations constructive in all instances, addressing each revision point-by-point. Edits and changes have been highlighted in blue color within the text.

General comments 

The study seems to be very underpowered for outcomes like injury incidence and length of rehabilitation - would be interesting to see the inputs used in your power calculation.

AR: Thank you for your comments, we understand the reviewer's concern. However, it should be kept in mind that to perform this type of study it is necessary that the soccer training and competition performed are similar in CG and GC, so in most cases they are performed in a single team. This aspect limits the recording of injuries and duration of recovery to the incidence of a single team. This same limitation and therefore similar samples have been used in other relevant studies as reported in L140. Statistical power has been calculated only for the sample size of participants. Injury variables are aspects that cannot be controlled by the research team; it is the result of the sport itself and the research itself. Taking this aspect into account, throughout the text no generalization of the results is made, but reference is made to the participating players. This aspect is already presented as a limitation of the study in the last paragraph of the discussion.

While the writing and grammar is not poor, it is inconsistent - consider seeking assistance with English and proofreading.

e.g L32 which players must be prepared to cope their optimal sport performance in each microcycle 

This does not make sense grammatically or otherwise. There are numerous occurrences like this throughout the manuscript. 

AR: Thank you for your comment. We have proceeded to send the text to proofreading to improve the quality of the document so that there are no inconsistent expressions.

The introduction makes a strong case for conditioning as an injury prevention strategy without really giving coverage to the multiple factors thought to contribute to athletic injury, including ‘chance’, foul play, environmental factors, fatigue, warm up, style of play, and of course strength and conditioning. Although it was reassuring to see that training load and injury risk was reviewed. 

AR: Thank you for your appreciation. We have added your contribution in L87-89 of the text.

No detail is provided on the time of season that the intervention was undertaken. Was this preseason? What are the implications of trying to build strength during a competitive phase. Periodisation is mentioned but not explored relative to your study. 

 AR: Thanks for the comment. In the original document, in L153 it is specified that it is performed in the middle of the season (The EG players conducted a 12-week mid-season intervention during the months of February to April, in addition to their soccer training). With respect to the periodization, Table 1 shows the periodization of the training protocol.

I am assuming that the football specific training was 4 x 105 min sessions = 420 mins /week?

 AR: Yes, this is correct. However, we have added in L142 for better understanding.

How did you find participants who had no previous neuromuscular injury?

AR: At the start of the study, one of the inclusion criteria was not to be injured, as indicated in the text. The squad consisted of 24 players, of whom the 20 selected had no previous injuries. Two goalkeepers and two players who were injured were not included in the study. This aspect was assessed by the club's medical services.

The figures depicting the resistance training exercises were helpful - I think C is commonly identified as leg extension and D as a leg curl (the femur is not really moving!)

AR:  We thank you for your appreciation. We agree and have proceeded to modify it in the text (L181), Figure 1 and Table 1.

Descriptions of isometric testing are not clear on the joint angle tested and how these were standardised/normalised across participants.

AR: Thank you for your comment. We have proceeded to improve the information and description of this section in the text between L297 – L305.

Table 3 - musculoskeletal structure rather than muscular? Are internal and external necessary when referring to ligamentous injuries.

AR: Thank you for your appreciation. We have modified the information in Table 3.

Figure 3 does not provide a scale or values 

AR: Thank you for your comment. Figure 3 shows the injury incidence (a) which is calculated as the number of injuries per 1000h of exposure and the injury burden (b) which is the days of absence per 1000h of exposure, both in the experimental group and in the control group. The specific results we found were as follows:

(Figure 3A, CG: 6.45 vs. EG: 1.15 injuries/1000 h of exposure, RR = 5.59, 95% CI 0.65-47.81), but significant differences (p < 0.001) in injury burden were reported (Figure 3B, CG: 145.77 vs. EG: 28.87 absence days/1000 h of exposure, RR = 5.05, 95% CI 3.27-7.79).

Table 4 - I wonder about the value of this table - it could probably be summarised in text.

AR: Thank you very much for your comment. It is one of our most important tables in our statistical analysis, and we consider it fundamental to understand well all the improvements that are obtained in the physical condition parameters that are analyzed in our study. Nevertheless, we appreciate your appreciation very much.

Discussion

Suggest care - you did not show that EG players experience fewer injuries statistically! There probably should be some commentary on the injury incidence in the EG group relative to norms for football - quite likely LUCK may have played a part and the nature of the injury likely had an influence on documented burden.

AR: Thank you for your comment. We believe that this statement is appropriate since it refers to the absolute number of injuries, an aspect that does not require statistical analysis. On the other hand, your assessment is very interesting and we have included it between L452 - L454.

A very useful finding that the additional training load appeared to be well tolerated. 

AR: Thank you for your comment. We totally agree, it is one of the most important findings of this study.

The resistance exercise used were very afunctional, so it does seem a bit of stretch to suggest that muscle architecture and balance ratios are a reason for a lower burden.

AR: Thank you very much for your observation. On the one hand, we have considered introducing exercises on guided machines to be able to work with such high loads and that there is no danger or any increase in intervertebral pressure. On the other hand, we have justified the great importance of muscular imbalance, especially between Q-H with studies of great relevance and a large sample of participants (ref. Nº 17 and Nº 68). In these studies, there is a clear association of injury risk in hamstrings if there is not a correct strength ratio between hamstrings and quadriceps (Q-H). Although we sincerely thank you very much for your appreciation.

I am not sure why the altered hip ABD/ADD balance was a surprise for the EG group given that you were targeting gluteus medius etc and ignoring the adductors. Given some arguments around groin injuries and symphysis pubis injury in football, I thought that this may have been explored further.  

AR: Thanks for the comment. Although no specific work was done to reduce the imbalance Abd add we were surprised by the Abd-Add imbalance increase. They are players with previous soccer training experience and we would have expected the Abd-Add imbalance to be maintained since we did not directly intervene in this.

Overall my view is that your discussion and conclusions are extending beyond what your data can support. In that respect I feel that the limitations need expansion and as noted the difficulty of isolating causation with injury needs to underpin discussions. The most significant finding for me was that extra RT did not adversely affect player wellbeing and perceived fatigue. 

AR: Thank you very much for this comment. We fully agree and that the proposed limitations have been added between L593 – L597, in addition to those already reflected.

We would like to thank you for all the comments you have made, because thanks to them this study will be significantly improved.

Finally, the 2 bibliographic references that were not cited due to an error with the Mendeley bibliographic program have been corrected.

Round 2

Reviewer 1 Report

Comments and Suggestions for Authors

Dear authors,

I would like to emphasize once again that your study is good in terms of design and the variables collected. However, due to the small number of cases, I try to ensure that the statistical analysis (sample size, application of inferential statistical methods) is presented correctly so that the reader cannot draw false conclusions from the data.

In my opinion, the calculation of the sample size using G*Power is incorrect. The central variables were collected at least at 2 time points (pre, post) and in 2 groups (EG, CG), therefore a repeated measures ANCOVA is the central method for the variables RPE, Hooper and physical fitness. For this reason, the calculated sample size is only correct if this method is selected in G*Power!

The revisions are good and support the quality of this study. I wish you continued success!

Kind regards

Author Response

Authors' Response (AR): We would like again to express our gratitude for the time you have devoted to review our manuscript and provide us with useful comments. We have ensured that the previous review has ameliorated the manuscript.

Thanks again for this comment. We agree with the reviewer in reference to calculate the repeated measures ANCOVA analysis. So, we have deleted the information about the power statistical analysis. We have tried to avoid make generalizations throughout the text.

The number of participants in each group was based on previous literature [41,42]. These studies used a similar design and methodology to our research. And these investigations recruited similar or smaller sample sizes as follows:

[41] Brull-Muria, E.; Beltran-Garrido, J.V. Effects of a Specific Core Stability Program on the Sprint and Change-of-Direction Maneu-verability Performance in Youth, Male Soccer Players. Int J Environ Res Public Health 2021, 18, doi:10.3390/ijerph181910116.

[42] Hung, K.C.; Chung, H.W.; Yu, C.C.W.; Lai, H.C.; Sun, F.H. Effects of 8-Week Core Training on Core Endurance and Running Economy. PLoS One 2019, 14, doi:10.1371/journal.pone.0213158.

In addition, in our study we have included higher number of variables in relation to perceived exertion training load and wellbeing state of participants. Therefore we consider that it is necessary to investigate these variables players who play in the same team to ensure that the training load is identical for both groups (control and experimental).

Thank you for your appreciation once again, we are sure that this observation will help readers not to misinterpret the analysis made.